# Tone Image Classification and Weighted Learning for Visible and NIR Image Fusion

**DOI:** 10.3390/e24101435

**Published:** 2022-10-09

**Authors:** Chan-Gi Im, Dong-Min Son, Hyuk-Ju Kwon, Sung-Hak Lee

**Affiliations:** School of Electronic and Electrical Engineering, Kyungpook National University, 80 Deahakro, Buk-Gu, Daegu 41566, Korea

**Keywords:** image fusion, deep learning, supervised learning, infrared image, visible image

## Abstract

In this paper, to improve the slow processing speed of the rule-based visible and NIR (near-infrared) image synthesis method, we present a fast image fusion method using DenseFuse, one of the CNN (convolutional neural network)-based image synthesis methods. The proposed method applies a raster scan algorithm to secure visible and NIR datasets for effective learning and presents a dataset classification method using luminance and variance. Additionally, in this paper, a method for synthesizing a feature map in a fusion layer is presented and compared with the method for synthesizing a feature map in other fusion layers. The proposed method learns the superior image quality of the rule-based image synthesis method and shows a clear synthesized image with better visibility than other existing learning-based image synthesis methods. Compared with the rule-based image synthesis method used as the target image, the proposed method has an advantage in processing speed by reducing the processing time to three times or more.

## 1. Introduction

To increase the reliability of images for CCTV, robot vision, and autonomous driving and to perform object recognition technology fully, it is necessary to provide error-free image information in any situation, including lighting, or space. A color camera uses an infrared cut-off filter, which blocks an infrared signal, to take a visible light image. Infrared light influx distorts color information, and in particular, since the amount of light in the daytime is highly intense, infrared light can easily saturate the image [1,2]. However, since the infrared image contains detailed information that is not expressed in the visible light image, the visibility of the image can be improved by the information fusion method. For example, in the case of fog and haze, since infrared rays have a stronger penetrability to particles than visible rays, more detailed information can be obtained from infrared images [3,4].

Visible and near-infrared (NIR) fusion algorithms have been developed by various methods, e.g., subspace-based methods, multi-scale transform, and neural networks. Subspace-based methods aim to project a high-dimensional input into a low-dimensional space or subspace. Low-dimensional subspace representations can be used to improve generalization. Principal component analysis (PCA) [5], non-negative matrix factorization (NMF) [6], and independent component analysis (ICA) [7] are mainly used for this method. Multi-scale transform-based methods assume images to be represented by various layers in different grains. In this method, the source image is decomposed to various levels and the corresponding layers are fused according to a specific rule. Then, the reconstructed image is acquired accordingly. Decomposition and reconstruction generally include wavelet [8], pyramid [9], and curvelet [10] methods. Neural network-based methods imitate the way the human brain processes neural information. This method has the advantages of good adaptability and fault tolerance [2,11].

In one of the subspace-based methods, low-rank fusion, Li et al. used low-rank representation to extract features, then reconstruct the fused image to use l1-norm and the max selection strategy [12]. Additionally, with one of the multi-scale transform-based methods, Laplacian–Gaussian pyramids and the local entropy-fusion algorithm, Vanmali et al. used Laplacian–Gaussian pyramids and local entropy to generate weight maps to control local contrast and visibility to produce fusion results [13].

Figure 1 shows the image fusion results using visible and NIR images through Vanmali method. As shown in Figure 1c, NIR images play an important role in image improvement because they contain many edge components of objects, and objects covered by clouds or fog can easily be identified with NIR rays.

However, such a rule-based synthesis method has a problem in that as the size of the image increases, the amount of calculation rapidly increases, resulting in a longer processing time. Therefore, it is not suitable for video processing for the object detection and classification of autonomous driving that requires high processing speed at high resolution.

With the rise of deep learning, various image synthesis methods based on deep learning have been proposed. The convolutional neural network (CNN) obtains image features by repeatedly applying filters to the entire input image and can synthesize images based on the features. This CNN-based image synthesis method does not take longer than the rule-based synthesis method even if the resolution of the input image increases. Therefore, the CNN-based synthesis method can improve the problem of the rule-based synthesis method [14]. In contrast, deep learning training requires a dataset containing a large number of captured images, but it is difficult to obtain sufficient training images via a deep learning training model for fusing visible and infrared images with a high degree of improvement for the same image scene. This problem is important in deep learning-based synthesizing methods because if the training dataset is insufficient, the training is not performed properly, which lowers the quality of the fused image.

To solve this problem, this paper presents a method for securing an effective training dataset and a classification method for obtaining a high-quality fused image. It provides a method for training and fusing the proposed model using DenseFuse [15], a CNN-based image fusion method. We propose a novel feature map fusion method that can improve the synthesis performance by tuning the training network in DenseFuse. Moreover, we suggest rapid color image synthesis with a proposed feature map synthesis method through color space channel separation in the fusion phase. At last, we conducted experiments to verify and evaluate the performance of the proposed method. The experimental results show that the proposed method is superior compared with several existing image fusion methods.

## 2. Related Works

### 2.1. Visible and NIR Image Fusion

The NIR wavelength band ranges from 800 to 1500 nm, and this wavelength band is higher than that of the visually recognizable band. In the NIR image, there is a region with strong contrast and sharpness of the object, and to use it, a technique for fusion after capturing both visible and NIR images may be applied. Visible images are advantageous for expressing color components when fusing images, whereas NIR images can be useful for expressing edges of objects and making accurate judgments because the boundaries and textures of the entire image are expressed in such images. Therefore, a clear and highly visible image can be reproduced by fusing an NIR image and a visible image, e.g., in fog or smokey conditions.

Among the image fusion technologies widely known and researched, there is a pyramid-based image fusion method [16]. Laplacian–Gaussian pyramids and the local entropy-fusion algorithm first make a weight map by measuring local contrast, local entropy, and visibility, then perform multiple resolution decomposition on visible and NIR images through the Laplacian–Gaussian pyramid, finally reconstruct image through the weight map. The Laplacian pyramid is a multi-scale processing algorithm that decomposes input images into multi-scale images [4]. To generate Laplacian pyramid, up-sample the Gaussian pyramid and subtract the previous level image of the Gaussian pyramid to obtain a differential image. The Gaussian pyramid is created by applying a Gaussian filter to an image to create a blurred image and then down-sampling the image to which the filter is applied. Then, the down-sampled image is up-sampled and subtracted from the image before down-sampling to generate the Laplacian pyramid image [17]. Thus generated Laplacian pyramid stores the differential images obtained by the Gaussian pyramid. Figure 2 shows a block diagram of Laplacian pyramids. G denotes Gaussian filter. For the Laplacian pyramid images in Figure 2, grayscale-normalized images were used because it is difficult to visually appreciate the original images.

Visible and NIR multi-resolution images generated through the Laplacian pyramid are synthesized for each layer with a weight map generated by measuring local contrast and local entropy to obtain a final composite image. It transmits the maximum information of local entropy to provide a composite image with improved visibility [13].

This method shows higher visibility in the haze region than other visible and NIR fusion methods and contains various details. However, there is a disadvantage that the processing time becomes longer as the resolution increases due to the rule-based synthesis method.

### 2.2. Deep Learning-Based Image Fusion

In recent years, deep learning methods are being actively applied in the field of image fusion, and it has been found that deep learning-based image fusion improves the time-consuming efficiency and the fusion effect [18,19]. Prabhakar proposed a CNN-based deep learning model that has a Siamese network architecture, which consists of two CNN layers in the encoder and three CNN layers in the decoder [14]. Each encoder requires two inputs, and the generated feature maps are fused using an addition strategy to obtain the final image at the decoder. Although this fusion method performs excellently well in terms of speed, the network structure is too simple to properly extract salient features. Additionally, there is a problem that useful information in the intermediate layer is lost when using only the result calculated from the last layer in the encoding network. To solve the information loss problem, DenseFuse is employed, which uses a set of convolutional layers with a cascade structure, called dense block, as a CNN model for fusing visible and infrared images [15]. Figure 3 shows the training architecture of DenseFuse. C1, DC1, DC2, DC3, C2, C3, C4, and C5 denote the convolutional block (convolutional layer + activation function). Dense block includes three convolutional blocks, and each block includes convolutional layers with 3 × 3 filter and ReLu activation function. Each layer included in the dense block can extract salient features by preserving deep features in the encoder network as the output is cascaded as the input of the next layer.

Additionally, DenseFuse suggested the possibility of improving the fused image by applying various fusion strategies in the fusion layer that synthesizes feature maps.

The addition fusion strategy synthesizes the salient feature maps obtained from the encoder through Equation (1). $\varphi_{i}\;\left(i=1,\;2,\;\cdots ,\;n\right)\;$indicates the feature map generated from each input and *n* indicates the number of inputs; *f* denotes a feature map in which the feature maps of each input are synthesized. This strategy has a speed advantage by simply adding to combine the outputs of previous layers [14].
(1)$$f\left(x,y\right)=\sum_{i=1}^{n}\varphi_{i}\left(x,y\right)$$

The l1-norm fusion strategy synthesizes a feature map by assuming that the l1-norm of the feature vector for each node represents the activity level. Each activity level map is generated from the feature maps by l1-norm as Equation (2), and an average operator is applied to the activity level map to obtain a weight map. In Equation (2), $\alpha_{i}$ (*i* = 1,2,⋯⋯, *n*) denotes an activity level map for the feature maps of each input.
(2)$$\alpha_{i}\left(x,y\right)=\sum_{i=1}^{n}\|\varphi_{i}\left(x,y\right){\|}_{1}$$

Finally, each feature map is multiplied by the generated weight map and added to obtain a fused feature map. Although this strategy gives better synthetic results than the addition fusion strategy in some situations, it is relatively slow [15].

Meanwhile, because DenseFuse lacks a pair of visible image and infrared image datasets, only the visible image dataset Microsoft Common Objects in Context [20] is used for training as unsupervised learning [15]. Despite receiving multiple inputs for fusion, training process is performed only with a single input so that the quality of the synthesized image cannot be guaranteed in the fusion phase.

## 3. Proposed Methods

In this paper, we propose a method for securing insufficient visible and NIR image datasets and a dataset selection method for effective training as well as proposing a training model and a fusion scheme to reconstruct an excellent fused image through the selected dataset. First, the target image to be used for learning is fused from the visible and NIR image datasets, RGB-NIR scene dataset [21] from EPFL, and sensor multi-spectral image dataset (SSMID) [22], through Vanmali’s fusion method [13]. Then, a dataset to be used for training is selected by comparing the luminance and variance values of the visible and NIR images. By training the proposed model through the selected dataset, high-quality visible and NIR-fused images are obtained.

### 3.1. Visible and NIR Image Pair Generation

From the visible and NIR image datasets, several 256 × 256 images, which are the input image size of the proposed model training step, are cut out at regular intervals and secured with only the luminance channel. Figure 4 shows a pseudocode of the proposed algorithm. In order to obtain images as many as possible, we acquired images cropped with as little overlap as possible through this pseudocode. Figure 5 shows how an input image is obtained from an image size of 1024 × 679 through the proposed algorithm. In this way, we were able to augment the visible and NIR images from 971 to 9883, respectively. The proposed algorithm is also applied to the previously fused target images to obtain the target images to be trained.

### 3.2. Local Tone-Based Image Classification

The difference image between the visible and NIR image pairs obtained by the proposed algorithm is obtained using Equation (3). In Equation (4), the visible and infrared images are denoted by $I_{vis}\;$and $I_{nir}$, respectively, and the difference image is denoted as $I_{diff}$. Using $I_{diff}$ obtained from Equation (3), the luminance average of $I_{diff}$ is obtained using Equation (4). In Equation (4), the average luminance value of $I_{diff}$ is denoted by $Dif_{l}$, and the size of $I_{diff}$ is given by NM. N and M represent the row of $I_{diff}$ and column of $I_{diff}$, respectively. Visible and infrared image selection of $Dif_{l}$ above a certain value is performed.
(3)$$I_{diff}\left(x,y\right)=\left|I_{vis}\left(x,y\right)-I_{nir}\left(x,y\right)\right|,$$
(4)$$Dif_{l}=\left[\sum_{x=0}^{N-1}\sum_{y=0}^{M-1}I_{diff}\left(x,y\right)\right]/NM,\;$$

The variance represents the effect of the square of the average luminance value of the difference image, as the difference in luminance between each pixel and the neighboring pixel in the image is greater. Therefore, it is easy to detect edges in the distributed image. Through the difference between the edges of each image, an image that is difficult to select due to the difference in the tone region can be selected with a clearer difference through the difference in the distributed image. Equation (5) shows an algorithm for obtaining a distributed image using a block-based average operator.
(5)$$\mu \left(x,y\right)=\left[\sum_{x^{\prime }=-r}^{r}\sum_{y^{\prime }=-r}^{r}I\left(x+x^{\prime },y+y^{\prime }\right)\right]/{\left(2r+1\right)}^{2},\;\;V\left(x,y\right)=\left[\left[\sum_{x^{\prime }=-r}^{r}\sum_{y^{\prime }=-r}^{r}I{\left(x+x^{\prime },y+y^{\prime }\right)}^{2}\right]/{\left(2r+1\right)}^{2}\right]-\mu {\left(x,y\right)}^{2},\;$$

In Equation (5), μ and V represent the mean and variance of the input image I, respectively, and r is the size of the block. We set r=2 in the proposed method.

From each variance image obtained using Equation (5), the difference image of the variance image is obtained through Equations (6) and (7). Similar to $Dif_{l}\;$in Equation (4), $Dif_{v}\;$is obtained using Equation (7), and the visible and infrared images in which the average value of the difference image of the dispersion image is greater than or equal to a certain value are classified.
(6)$$V_{diff}\left(x,y\right)=\left|V_{vis}\left(x,y\right)-V_{nir}\left(x,y\right)\right|,$$
(7)$$Dif_{v}=\left[\sum_{x=0}^{N-1}\sum_{y=0}^{M-1}V_{diff}\left(x,y\right)\right]/NM,$$

Figure 6 shows the variance difference and luminance difference of the augmented visible and NIR images as histograms. The difference values of the images tended to be concentrated in specific values, and we determined the classification criteria by visually analyzing the images based on the specific values. In Figure 6, the blue stars indicate the values referenced by the analysis, and the values corresponding to the red stars mean the criteria values because the difference between the visible image and NIR image is visually clear. The luminance difference was relatively easy to visually identify the difference in the image, and the value corresponding to the top 36.4% was set as the criterion value. In the case of variance difference, the value corresponding to the top 94.1% was set as the criterion value in order to maintain the number of images classified by luminance difference as much as possible.

Through this classification, we selected a total of 3431 visible and NIR images for training, respectively. Additionally, among the images not used for training, 30 image pairs were used as the validation set, and 26 image pairs were used as the testing set.

### 3.3. Weighted Training

Figure 7 shows the learning structure of the proposed model. The proposed model divides the channel according to the input so that the synthesis structure and the learning structure are matched. Here, the numbers at the bottom of each convolutional layer represents the size of the filter and the number of input and output feature maps. The proposed model consists of an encoder, fusion layer, and a decoder. Inputs go into channel 1, composed of C11, DC11, DC21, and DC31 and channel 2, composed of C12, DC12, DC22, and DC32 in the encoder network. For each input to generate a feature map, convolutional layers having the same structure in each channel are computed in parallel. At this time, DC11, DC21, and DC31 (or DC12, DC22, and DC32) have a cascade structure, so that useful information of each convolutional layer can be learned without much loss. The generated feature maps are fused together after being multiplied by appropriate weights in the fusion layer. Weights are determined as the model is trained.

The proposed method learns weights to be multiplied by feature maps for fast image synthesis and optimal synthesized image quality. The learned weights are multiplied by each feature map and synthesized by the addition method. Equation (8) and Figure 8 show the process of multiplying and merging weights from the generated feature map. $w_{1}$ and $w_{2}$ denote multiplied weight, and φ indicates a feature map generated from each channel. f denotes the synthesized feature map and m denotes the index of the feature map (m=1, 2, ⋯, 64).
(8)$$f^{m}\left(x,y\right)=\sum_{n=1}^{2}w_{n}\times \varphi_{n}^{m}\left(x,y\right),$$

A final image is generated through a total of four convolutional layers in the decoder network. The encoder, fusion layer, and decoder are trained to minimize the loss function by comparing the final image generated to learn the target image with the acquired target image. In Equation (9), the loss function is denoted by L. In Equation (10), the pixel loss function is denoted by $L_{p}.$ In Equation (11), the structural similarity (SSIM) loss function is denoted by $L_{ssim}$ with the weight λ [14].
(9)$$L=\left|\lambda \times L_{ssim}\right|+L_{p},$$
(10)$$L_{p}=\|O-T{\|}_{2},$$
(11)$$L_{ssim}=1-SSIM\left(O,T\right),$$

Here, O and T represent an output image and a target image, respectively. The pixel loss function is the Euclidean distance between the output image and the target image. SSIM(·) indicates the SSIM operator and compares the SSIM between two images [23]. In training phase, since $L_{p}$ has a value about three orders of magnitude larger than $L_{ssim}$, and the weight of $L_{ssim}$ can be increased by multiplying λ. In the proposed method, the λ is set to 1000, which can reduce time consumption in the learning phase [15].

### 3.4. Image Fusion

Figure 9 shows the fusion method of the proposed model. First, an image to be fused is divided into luminance channel l and color channels a and b through CIELAB color space conversion. The CIELAB color space is based on the color perception characteristics of human vision, has excellent color separation, and is widely used in image tone mapping models to preserve and compensate color components [24]. The color channel $a_{vis},b_{vis}\;$of the visible image is preserved and used as the color channel of the fused image. Separate $l_{vis,}\;l_{nir}$ of the visible and NIR image used as inputs are fed into the input of each channel of the learned encoder network. The feature map output from the output of each channel are fused at the ratio trained from the fusion layer, and the fused feature map enters the input of the decoder network to obtain a fused luminance image $l_{fused}$. Finally, $a_{vis},b_{vis}\;$are merged into $l_{fused}$, and a color image is obtained through RGB color space conversion.

## 4. Experimental Results

To compare the training results of the proposed methods, after training the dataset under various conditions through the proposed model, various visible and NIR images were fused, and the similarity between the target image and the resulting image was compared through SSIM values. Table 1 shows the average of the SSIM values between the fused image and the target image after fusing 26 images from the model trained for each dataset. Lum_var_above refers to a dataset selected from images in which the average values of luminance and variance difference used in the proposed model are above each criterion value. Lum_above refers to a dataset selected by considering only the luminance difference value, and Lum_below refers to a dataset in which the luminance difference average value is less than the criterion value. Finally, Not_considered refers to a dataset that does not consider luminance and variance difference values. From the results, it can be seen that the model trained with Lum_var_above has the highest similarity to the target image.

Additionally, in order to check the effect on image fusion according to the ratio at which the feature map is fused in the fusion layer, images are obtained through the pro-posed model by varying the fusion ratio, and to check whether the performance of the proposed method is excellent, the image fusion methods of Lowrank [12], DenseFuse [15], and Vanmali [13] were compared with the image quality metrics.

Table 2 shows the average of the values obtained through the quality metrics by acquiring a total of 26 visible and NIR-fused images for each fusion method. Weighted_addition1 is a model in which only the weights multiplied by the infrared feature map are learned in the fusion layer of the proposed method, and Weighted_addition2 is a model in which the weights multiplied by the infrared feature map and the visible feature map to be fused are trained, respectively. Additionally, there is a model fused by the addition strategy without training the weights to be multiplied by the feature map. Both LPC [25] and S3 [26] evaluate the sharpness of the image. FMIpixel [27] indicates how much information of two input images to be fused is preserved. Qabf [28] evaluates image quality. The cross entropy [29] shows how similar the source image is to the fused image using information contents. The average gradient [30] can reflect detail and textures in the fused image. The larger the average gradient means that the more gradient information is contained in the fused image. The edge intensity [31] represents the quality and clearness of the fused image. The spatial frequency [32] metric indicates how sensitive and rich in the edges and textures are according to the human visual system.

Here, Weighted_addition2 has the best score value for the four quality metrics (LPC, FMIpixel, average gradient, and edge intensity) and the second-best value for the three other metrics (S3, Qabf and spatial frequency). Thus, it can be confirmed that among the proposed methods, Weighted_addition2 shows slightly better results for image fusion. Overall, the proposed method shows an insignificant improvement of 1% to 2% compared to the existing method in LPC and FMIpixel metrics, but shows improvement in quantitative metrics of 5% to 22% in S3 and Qabf metrics. Additionally, the proposed method shows 39% improvement in the cross entropy compared to the existing method, and the proposed method shows improved performance of 5% to 18% in average gradient, edge intensity, and spatial frequency. It can be seen that the proposed method acquires a clear, high-quality image with less distortion compared to other fusion methods.

## 5. Discussion

The proposed model, the existing visible and infrared image fusion method, and several images were fused and evaluated for comparison. Figure 10, Figure 11, Figure 12, Figure 13, Figure 14, Figure 15 and Figure 16 show the input visible and NIR images and the resulting images. It can be seen that Figure 10e,d. the target image, contain the detailed information of the image better in the shaded area compared to Figure 10b,c. Particularly, in Figure 10b, the edge of the shaded area is hardly expressed. In Figure 11c, it can be seen that the detail does not appear well in the distant mountain part. Particularly, in Figure 11b, it can be seen that not only the detail of the mountain but also the overall quality of the image has deteriorated. In contrast, in the proposed method, Figure 11e shows a clear, high-quality image. As for the composite result of Figure 12b,c, the expression of the boundary between the mountain boundary and the trees and buildings is inferior. In contrast, in the proposed method, Figure 12e learns the target image, Figure 12d, so that the boundary expression is good and the visibility is excellent.

In Figure 13d,e, the boundaries between the trees are clear and the detail of the leaves is superior to Figure 13b,c. The proposed method, Figure 14e and the target image, Figure 14d, have information on distant mountains that are not visible in Figure 14b,c. In addition, the proposed method provides a clearer image even in the details of the grass part. In the case of Figure 15b, the buildings beyond the glass that is concealed in the visible image can be seen clearly, but the detail of the tree part that is detectable in the visible image is inferior. Figure 15c has an overall blurry image. However, in Figure 15e, not only the trees but also the buildings beyond the glass are clear. In Figure 16b,c, the boundary between the tree part and the structure is blurred, so the structure cannot be clearly distinguished. In Figure 16d,e, the boundary is clear and the image quality is excellent, so the structures can be distinguished well.

Table 3 shows the processing time comparison compared to the Vanmali fusion method, which is the target image fusion method of the proposed model, and the image fusion processing time of the proposed method. For each method, images were fused 10 times by each resolution, and the average processing time was calculated and compared with those of the other methods. As the resolution increases, it can be seen that the deep learning-based fusion method has a significantly faster processing speed than the rule-based fusion method. Both methods were implemented with NVIDIA RTX 2060 GPU and i5-6500 CPU as a common PC and NVIDIA RTX 3090 GPU and i9-10980XE CPU as a high-performance PC, respectively.

## 6. Conclusions

In this paper, we propose a method for reducing the processing speed while preserving the excellent synthesis quality of the rule-based image synthesis method using the deep learning-based visible light and near-infrared image synthesis method. The proposed method learns the excellent detail expression of the rule-based image synthesis method by presenting a data set acquisition method and a classification method for effective learning.

The proposed method has been compared with several existing synthesis methods through quantitative evaluation metrics, and the results of the metrics have been improved by 5% to 22% in the S3, Qabf, average gradient, edge intensity, and spatial frequency metrics. In particular, the proposed method shows 39% improvement in cross-entropy compared with the existing methods, and in the comparison of visibility through the result image, the proposed method not only shows the excellent resulting image but also shows the synthesized image quality equal to or higher than the target image. In addition, by using a deep learning model, the amount of computation is reduced, and the processing speed is three times faster than the target image synthesis method. This means that it can be considered as a method more suitable for video synthesis than existing synthesis methods.

## Figures and Tables

**Figure 1 entropy-24-01435-f001:**
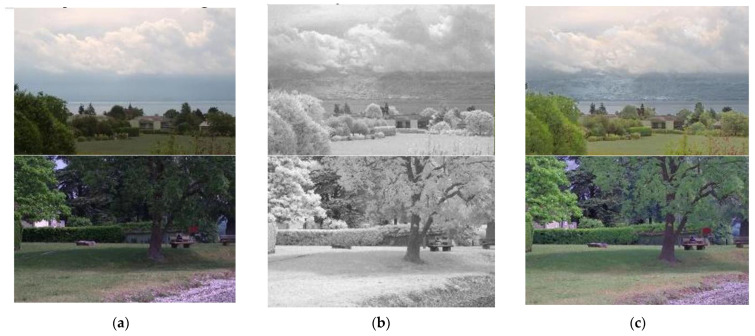
Visible light and NIR input and result images: (**a**) visible image, (**b**) NIR image, and (**c**) fused image.

**Figure 2 entropy-24-01435-f002:**
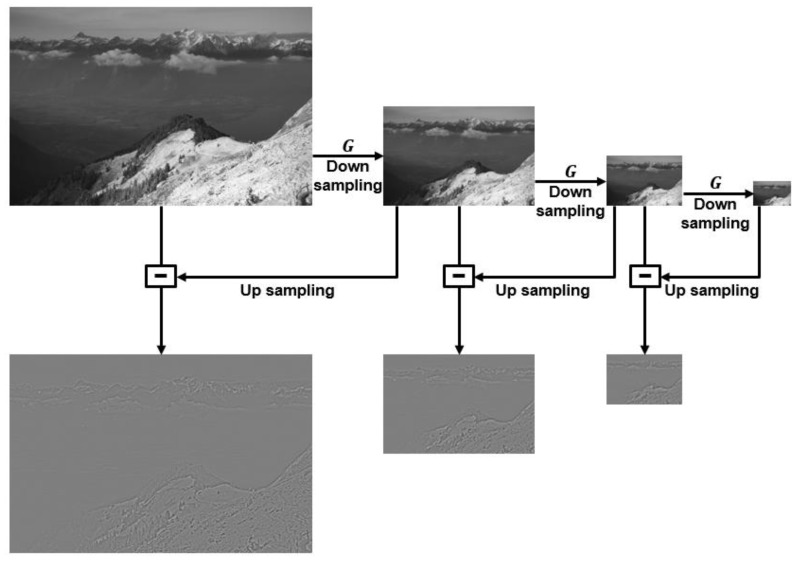
Block diagram of Laplacian pyramid.

**Figure 3 entropy-24-01435-f003:**
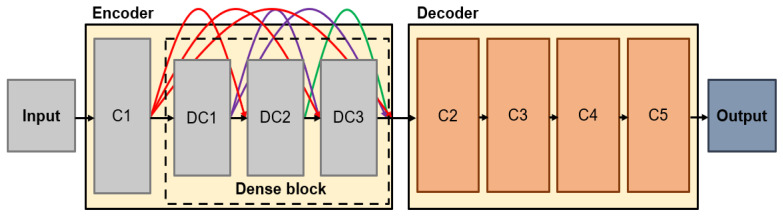
DenseFuse architecture for training.

**Figure 4 entropy-24-01435-f004:**
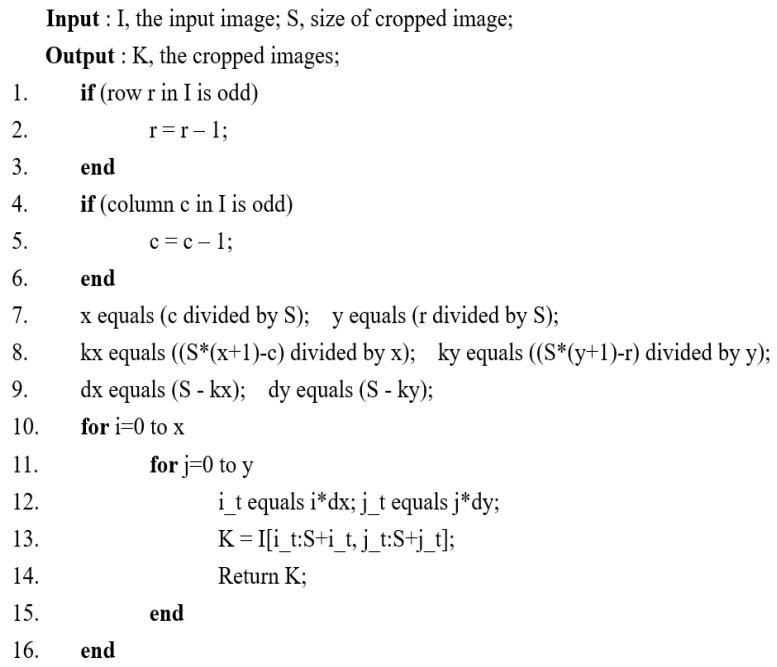
Pseudocode of the proposed algorithm for image cropping.

**Figure 5 entropy-24-01435-f005:**
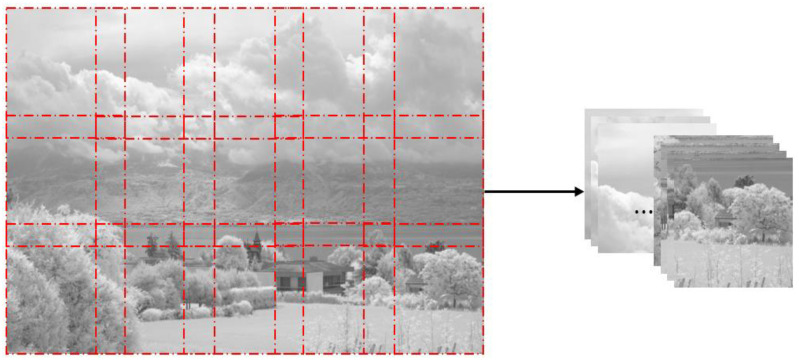
Image acquisition by the proposed algorithm for image cropping.

**Figure 6 entropy-24-01435-f006:**
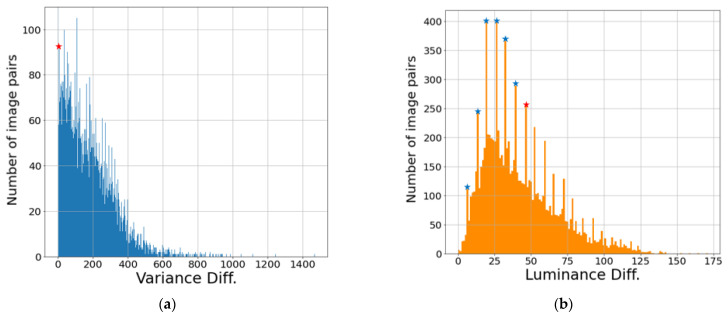
Difference between the augmented visible and NIR images histograms: (**a**) variance difference histogram and (**b**) luminance difference histogram.

**Figure 7 entropy-24-01435-f007:**
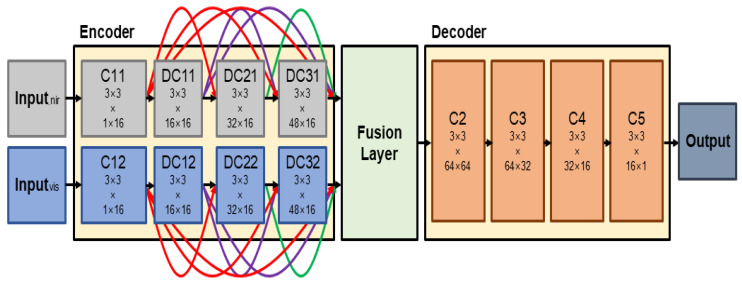
Learning architecture of the proposed model.

**Figure 8 entropy-24-01435-f008:**
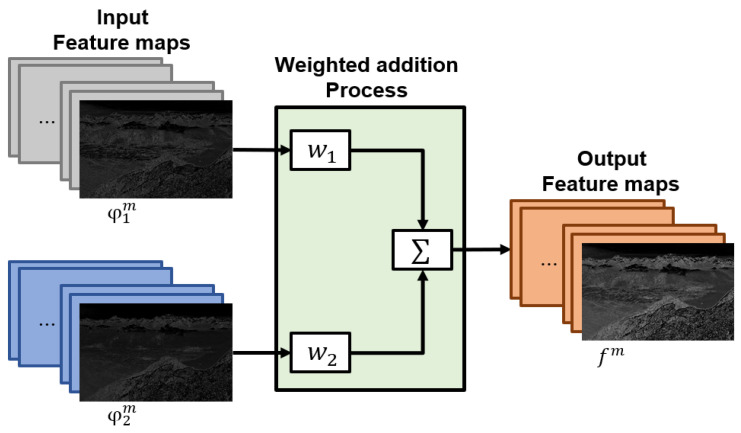
Feature maps fusion scheme of the proposed model.

**Figure 9 entropy-24-01435-f009:**
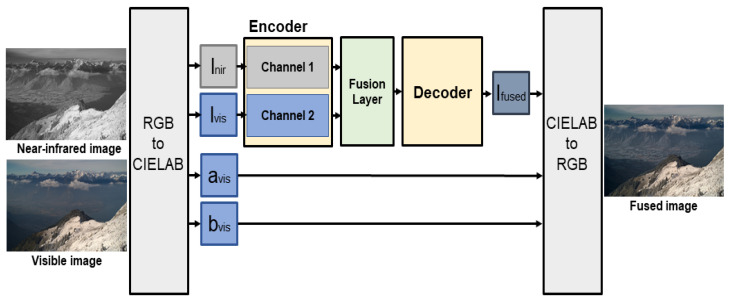
Fusion scheme of the proposed model.

**Figure 10 entropy-24-01435-f010:**
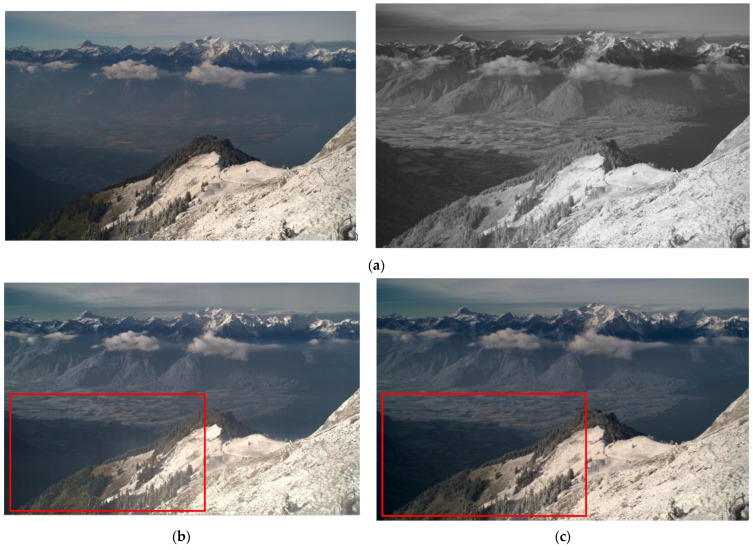
Input and result images (1): (**a**) visible and NIR images, (**b**) low rank, (**c**) DenseFuse, (**d**) Vanmali method (target), (**e**) proposed model.

**Figure 11 entropy-24-01435-f011:**
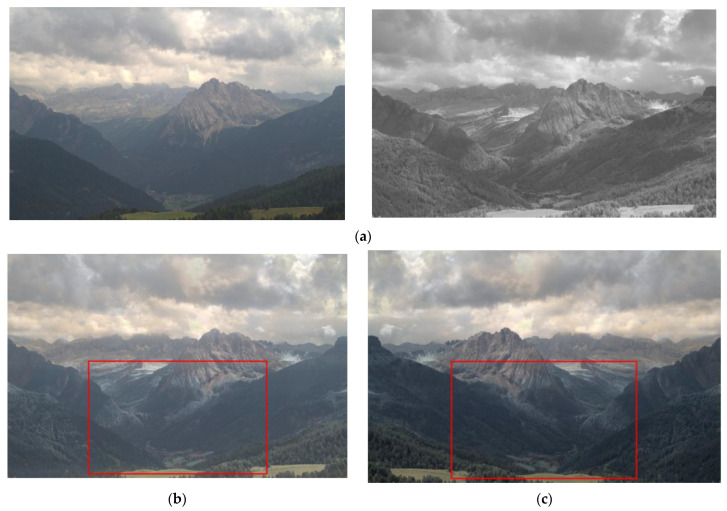
Input and result images (2): (**a**) visible and NIR images, (**b**) low rank, (**c**) DenseFuse, (**d**) Vanmali method (target), (**e**) proposed model.

**Figure 12 entropy-24-01435-f012:**
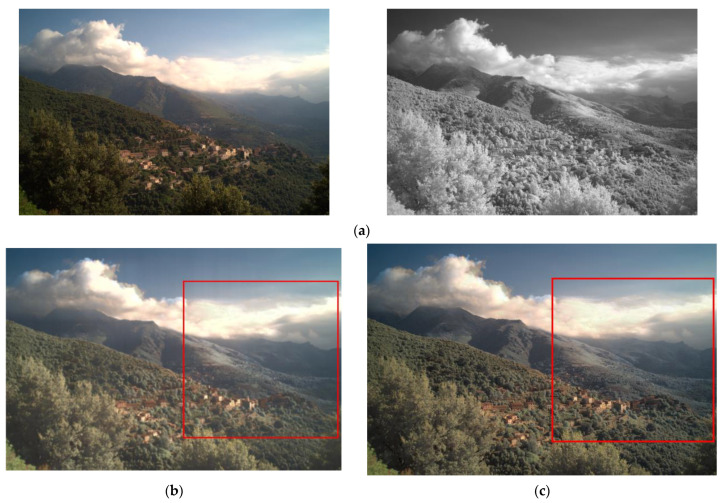
Input and result images (3): (**a**) visible and NIR images, (**b**) low rank, (**c**) DenseFuse, (**d**) Vanmali method (target), (**e**) proposed model.

**Figure 13 entropy-24-01435-f013:**
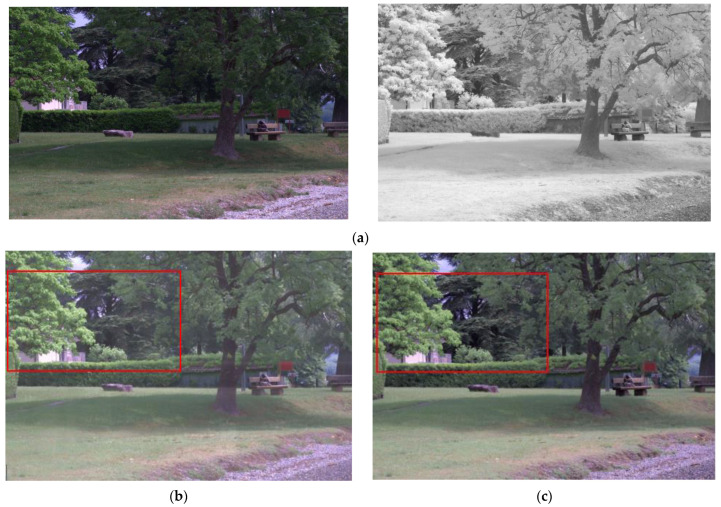
Input and result images (4): (**a**) visible and NIR images, (**b**) low rank, (**c**) DenseFuse, (**d**) Vanmali method (target), (**e**) proposed model.

**Figure 14 entropy-24-01435-f014:**
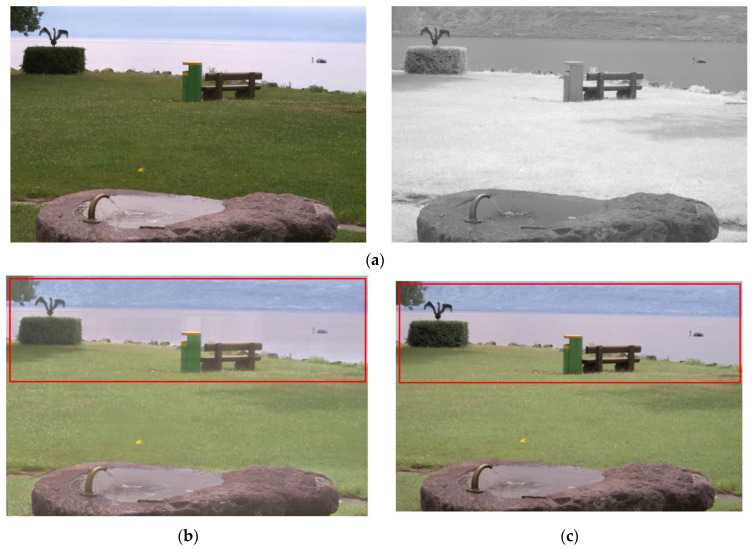
Input and result images (5): (**a**) visible and NIR images, (**b**) low rank, (**c**) DenseFuse, (**d**) Vanmali method (target), (**e**) proposed model.

**Figure 15 entropy-24-01435-f015:**
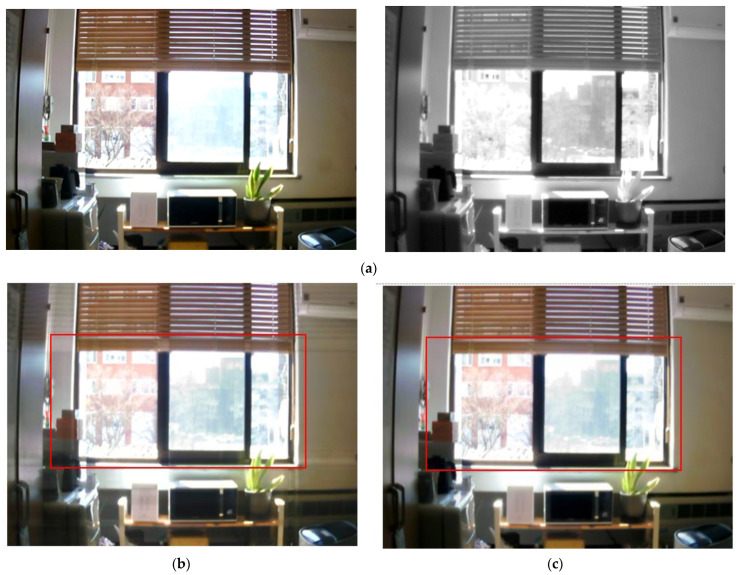
Input and result images (6): (**a**) visible and NIR images, (**b**) low rank, (**c**) DenseFuse, (**d**) Vanmali method (target), (**e**) proposed model.

**Figure 16 entropy-24-01435-f016:**
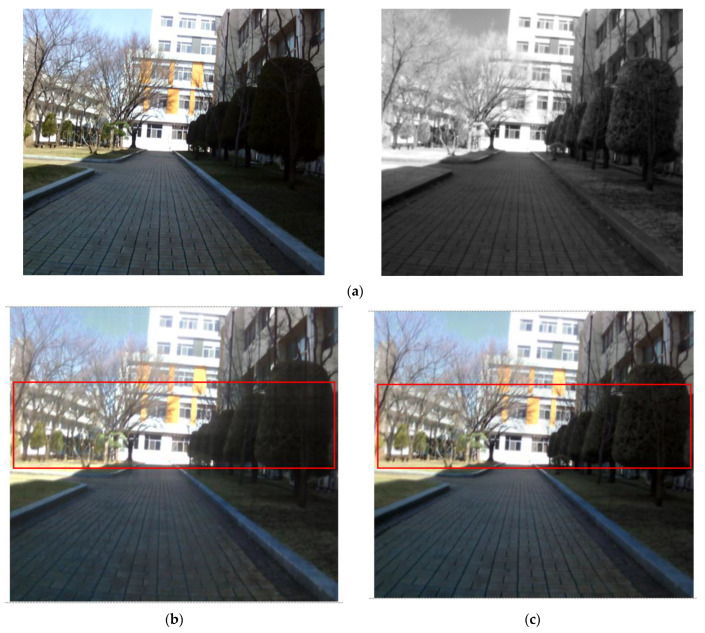
Input and result images (7): (**a**) visible and NIR images, (**b**) low rank, (**c**) DenseFuse, (**d**) Vanmali method (target), (**e**) proposed model.

**Table 1 entropy-24-01435-t001:** Comparison of SSIM values between target images.

Dataset	SSIM Value
Lum_var_above	0.93756
Lum_above	0.93468
Lum_below	0.92303
Not_considered	0.92634

**Table 2 entropy-24-01435-t002:** Image quality metrics score.

		LPC [25]	FMIpixel [27]	S3 [26]	Qabf [28]	Cross Entropy [29]	Average Gradient [30]	Edge Intensity [31]	Spatial Frequency [32]
Lowrank [12]		0.93609	0.86278	0.1458	0.49425	0.6754	3.94558	39.1852	12.5462
DenseFuse [15]		0.92951	0.87916	0.15111	0.57522	0.81813	4.10702	40.6721	12.1937
Vanmali [13]		0.94004	0.87817	0.1612	0.57579	0.84289	4.55613	45.4734	13.5013
Proposed model	Addition	0.94096	0.88261	0.15504	0.60661	0.94073	4.575	46.2832	12.9968
	Weighted_addition1	0.94107	0.88218	0.15567	0.60117	0.94065	4.58824	46.3161	12.9534
	Weighted_addition2	0.94284	0.88334	0.15792	0.60501	0.92142	4.62718	46.6067	13.2798

**Table 3 entropy-24-01435-t003:** Processing time comparison.

Processing Time (S)				
Image Resolution	Common PC		High-Performance PC	
	Vanmali Model	Proposed Model	Vanmali Model	Proposed Model
580 × 320	0.1966	0.0745	0.1089	0.0365
1024 × 679	0.3345	0.1494	0.2007	0.0676
1920 × 1080	0.7648	0.3085	0.4063	0.1349

## Data Availability

Not applicable.

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
