# Peer review of "Tone Image Classification and Weighted Learning for Visible and NIR Image Fusion"

_entropy, 2022, doi:10.3390/e24101435_

Round 1

Reviewer 1 Report

The paper is well constructed and clearly written. However, there are minor mistakes that should be improved before the publication:

- Figure 3. is not informative or not relevant. Authors should deeply consider removing or replacing the figure since it gives no information.

- in sec. 3 we can read "To check the training performance". However, there is no information about the training process - it has to be improved.

- in sec. 3 again, we can read "average values [...] are above 13 and 47, respectively" - where do these values come from? It has to be mentioned.

- in Table 2.the references to compared method could be useful.

- Table 2 and Figure 9. give the same portion of information - reconsider removing one of the redundant elements.

- for processing time comparison it'd be nice to get the specification of the testing device (times will be different on powerful GPU and just a common PC).

- I'd suggest also the sources revision - only one article from 2022. The references could be more up to date.

Author Response

We attached the reply letter. Thank you so much.

Reviewer 2 Report

The authors need to improve the structure of the manuscript.

The authors build on existing methods and propose a novel combination. However, they do not explain clearly which parts of the presented method is their own.

The structure  and number of the training, validation and test set is not presented clearly to reproduce the results.

The learning architecture is based on a autoencoder architecture. It has a lot of similarity with the architecture of DenseFuse work.

The weight combination to generate the fused image also has similarity with the work of Vanmali et al.

Please explain clearly in which aspects your proposed method uses other works and which parts are your own.

The sources of the datasets used (RGB-NIR,SSMID) are not provided (Section 2.3).

You need to explain in detail how many images are selected from each dataset, then when you apply your selection procedure, how many images you finally get.  Then describe how many pair of images (visible and NIR) are used for training, validation and testing.

A major assumption in the paper, that is not justified, is that the target/reference image is generated using Vanmali et al method. Why is that? Is this method the state of the art? Are there other methods to generate reference images?

Other specific comments:

1. The end of the Introduction section needs to clearly explain your contributions.

Also explain the structure of the document.

The last paragraph of section 1 (lines 76 - 81) does not explain well the contribution.

"It provides a method for training and fusing the proposed model using DenseFuse [15]..."

To which extent do you use DenseFuse?

What are your improvement?

lines 80 - 81:

"The experimental results show that the proposed method is superior compared to the existing image fusion method."

The above statement is not clear, you should compare with several existing fusion methods, not just one.

Actually in your experiments you compare with several works.

2. Line 106:

"... it is difficult to visually confirm the original images"

confirm -> appreciate

3. There are 2 subsections in Section 2, Materials and Methods, that should be moved to a Related Works Section, after the Introduction.

This because subsection 2.1, an subsection 2.2 describe related works, not your method.

4. Line 165 mentions Vanmali et al work but does not have a reference to them.

5. Lines 173-174 mentions that cropped images were extracted without overlapping, but Figure 5, left side presents overlapping lines. Please clarify.

6. Please complete the captions of Figures 4 and 5. For instance, the caption of Figure 4 should read:

"Pseudocode of the proposed algorithm for image cropping"

7. The title of section 2.3.1, can be modified to:

Data augmentation, Visible and NIR image pair generation

8. Line 190, the meaning of N and M is not described.

9. Equation (5), please check the expression for the block based Variance.

10. The variance image is not defined.

Maybe you can complement equations (5) to define

mu(x,y) = ...

V(x,y) = ...

11. Line 211

number -> numbers

12. lines 232 - 239, you should include a reference to the work of DeepFuse, equations (9) to (11) are the same as in that paper.

Please explain why you chose lambda = 10**3, in equation (9).

13. Section 2.3.4 Image Fusion, you fusion scheme and reconstruction also has similarity with the work of Valmani et al., please explain the similarities and differences.

14. Section 3, Experimental Results, lines 285 - 287, there are several image quality metrics that are not defined such as LPC, S3, FMIpixel, Qabf, please provide at least a reference to check these metrics.

15. Section 3, Experimental Results, lines 279 - 280, mentions that Table 2 and Figure 9, only show the comparative results for processing 26 visible and NIR image pairs.

Does your test set only has 26 image pairs?

16. Line 328, please improve this sentence, to avoid repeating words:

"...but the detail of the tree part that  is visible in the visible image is inferior."

17. Lines 309 - 362, should be the Discussion section.

18. Line 362, the section named Discussion appears to be the Conclusion section.

19. Please improve this sentence to avoid repeating words, and also to highlight that you compared with several methods not just with one.

"The proposed method has been compared with the existing learning-based synthesis 369 method through quantitative evaluation metrics"

20. The related works section should be improved explaining other recent related works and some review papers like:

From pyramids to state-of-the-art: a study and comprehensive comparison of visible–infrared image fusion techniques. 2020. https://ietresearch.onlinelibrary.wiley.com/doi/10.1049/iet-ipr.2019.0322

Sun, C.; Zhang, C.; Xiong, N. Infrared and Visible Image Fusion Techniques Based on Deep Learning: A Review. Electronics 2020, 9, 2162. https://doi.org/10.3390/electronics9122162

Author Response

(The authors gave the same response as above.)

Round 2

Reviewer 2 Report

The authors have addressed all the observations.

The manuscript has greatly improved. 

The manuscript has one minor error. Please correct it.

1.

Please check the caption of Figure 6.

The caption for Figure 6 (a) should read: luminance difference histogram.  The caption for Figure 6 (b) should read: variance difference histogram.